# A Study on Teachers' Continuance Intention to Use Technology in English Instruction in Western China Junior Secondary Schools

Yi Xie [1], Azzeddine Boudouaia [2,*], Jinfen Xu [1,*], Abdo Hasan AL-Qadri [3], Asma Khattala [4], Yan Li [2] and Ya Min Aung [5]

[1] School of Foreign Languages, Huazhong University of Science and Technology, Wuhan 430074, China
[2] College of Education, Zhejiang University, Hangzhou 310058, China
[3] School of Humanities and Education, Xi'an Eurasia University, Xi'an 710199, China
[4] Department of English Language and Literature, Mohamed Lamine Debaghine Setif 2 University, Setif 19000, Algeria
[5] Department of Teacher Education, Yankin Education College, Yangon 110-71, Myanmar
* Correspondence: azzeddine-link@hotmail.com (A.B.); xujinfen@hust.edu.cn (J.X.)

**Abstract:** This study aimed to investigate the factors that affect the continuance intention to use technology among English teachers in China, mainly by examining the direct effects of help seeking, interest, effort regulation, growth mindset, facilitating conditions, perceived usefulness, and perceived ease of use on continuance intention (CI), and the indirect effects the above factors have on continuance intention through self-efficacy. The study sample comprised 459 English language teachers from junior secondary schools in different regions in Western China. A questionnaire that involved the above variables was used, and it was validated using exploratory factor analysis and confirmatory factor analysis. The results revealed significant direct effects of help seeking, effort regulation, growth mindset, facilitating conditions, perceived usefulness, and perceived ease of use on the continuance intention to use technology. However, the results showed that interest did not have a direct effect on the continuance intention to use technology. The findings also demonstrated that growth mindset, interest, effort regulation, help seeking, and perceived usefulness did not indirectly affect the continuance intention to use technology through self-efficacy. Nevertheless, the findings indicated that facilitating conditions and perceived ease of use did have an indirect effect on the continuance intention to use technology through self-efficacy. In light of these findings, some suggestions and recommendations were presented.

**Keywords:** continuance intention; instruction; self-efficacy; teachers; technology





## 1. Introduction

English instruction has mushroomed worldwide over the past decades due to its significance for national development. As a consequence of the options that it provides for teachers and students, the teaching of this language as a foreign language has advanced simultaneously with the ever-accelerated advancement of technology. In view of this, it has been noticed that the incorporation of technology into English instruction in China has substantially increased. Every English as a foreign language (EFL) reform attempt must include the use of technology [1,2].

Technology has been seen as an e-learning tool for democratizing classroom communication, and this has afforded some English teachers with new possibilities to encourage Chinese students to utilize English [3,4]. Now, teachers have a greater variety of implementation method options (e.g., online and blended, in addition to face-to-face) [5,6]. In this sense, the continued use of technology is crucial for sustaining academic success in an EFL environment. The continuance intention is essential because it provides consistency and

sustainability for the advancement of teaching and learning. Despite its significance, there are few studies examining the continuance intention (CI) of technology use among EFL instructors [7,8]. An increasing amount of research and attention is being paid to the role that technology may play in EFL classrooms [9–11]. The use of technology in the classroom has been shown to be notably useful for the instruction and improvement of fundamental language abilities, including listening, speaking, reading, writing, grammar, vocabulary, and pronunciation [8,9,12–14]. However, teachers of languages (as opposed to subjects like physics and math) are often slower and less effective in technology integration due to their own educational backgrounds [15–17]. In this regard, it seems necessary to explore continuance intention in the EFL context.

Furthermore, there is a shortage of studies that investigate the continuance intention to use technology in instruction, and a need for future research [18]. For example, some previous studies focused more on the acceptance and adoption of multimedia online learning [19]. Other studies have investigated the continuation of technology use in terms of technostress and attitudes [20], perceived pedagogical impact and user interface quality [21], satisfaction in the blended learning context [22], intrinsic and extrinsic work motivation and occupational stress (i.e., burnout and technostress, which have been examined in tandem) [23], and perceived convenience and curiosity [24]. Previous research results suggest that other variables may affect the continuance intention to use technology [25,26]. These factors include learning behaviors, perceptions, motivational beliefs, and facilitating conditions [27–30]. More particularly, some studies have explored growth mindset (GM) with technology use [31,32], interest (IN) with ICT [33], facilitating conditions (FC) and help seeking (HS) with mobile learning and e-learning [25,26], and perceived usefulness (PU) and perceived ease of use (PEU) with general intentions to use technology [34,35]. However, to the best of our knowledge, there is a shortage of investigations into the impact of these factors on the continuance intention to use technology in an EFL context [7]. Moreover, with the advent of technology in education, English teachers form their own beliefs about their abilities in using technology, which may have a role in determining their intention. There is a strong correlation between teachers' levels of self-efficacy (SE) and the quality of the learning environments they foster, suggesting that instructors' confidence in their own abilities to implement new strategies may have a significant influence on student achievement [36,37]. In studies of teachers' propensity to accept new forms of technology, self-efficacy has been shown to be an essential variable to examine [38–40]. Self-efficacy may reduce the impact of several factors on continuance intention since instructors who have high levels of it are more likely to stick with their goals despite setbacks and to find creative solutions to problems that arise [41,42]; hence, their self-efficacy can mediate the effects of different factors on their continuance intention. Therefore, this study introduces growth mindset, interest, facilitating conditions, effort regulation, help seeking, perceived usefulness, and perceived ease of use as factors that can affect teachers' continuance intention to use technology in instruction in the Chinese context and from the teacher's perspective. In addition, this study introduces SE as a mediator variable that can mediate the effects of the above factors on continuance intention.

Despite the researchers' review of previous studies on teachers' intentions to keep using technology in the classroom, there was a lack of high-quality empirical studies on teachers' intentions to keep using technology in the classroom, which included the perspective of English language teachers and the mediating effects of self-efficacy. Our study set out to fill that void. This research aimed to better understand the variables that influence English instructors' continuance intention for adopting new technologies in the classroom. Specifically, this research aimed to examine the effects of help seeking, interest, effort regulation, growth mindset, facilitating conditions, perceived usefulness, and perceived ease of use on continuance intention, and the role of self-efficacy in mediating the relationship between these factors and continuance intention. With the objective of enhancing the quality of EFL instruction and learning, this study may aid policymakers in making better informed choices on the deployment of different resources to support teachers' professional

development in the area of classroom technology usage. It is conceivable that this study may offer cutting-edge research on the pressing issue of the continuance intention to use technology, which is of great concern to schools and governments worldwide. These latest results not only verified the findings of earlier studies, but also improved our knowledge of how EFL instructors interact with and use technology. This has far-reaching implications for the globalization of English in the context of foreign language instruction.

## 1.1. Theoretical Background

The technology acceptance model (TAM) was first put forward by Davis and his colleagues [43]. The TAM has always been the most frequently applied model for depicting technology acceptance in the domain of education [44]. The model proposes that the intention to utilize technology tools is influenced by two perceptions, i.e., the perceived usefulness and the perceived ease of use of the tools [45]. Later, the scope of the TAM was expanded by Rauniar (2014) [46], with other factors being included, such as facilitating conditions that highlight environmental characteristics. The above-mentioned perceptions and the factor of facilitating conditions have been studied and shown to be useful in predicting people's acceptance and utilization of modern information technology.

In recent years, an increasing number of researchers have investigated how motivational beliefs shed light on perceptions and technology acceptance [47]. The power of motivational beliefs is that they highlight teachers' general beliefs about technology use on the basis of previous experience [48]. These motivational beliefs can have a profound effect on people's perceptions about the utilization of technology tools and applications.

The expectancy–value theory, one of the most powerful motivational theories, argues that people's expectancies about the possibilities of success (e.g., self-efficacy) and subjective task values (e.g., utility values, playfulness, and cost) tend to determine their initiation and perseverance [49]. The more an individual is convinced that s/he can perform a task well, the more enjoyment s/he will obtain in performing the given task, and the less pessimistic s/he will be in the process of performing that activity. All these factors will be conducive to a higher level of acceptance of that task. In terms of technology application, positive personal traits such as self-efficacy will contribute to positive intentions.

In addition, in the process of defining continuance intention, a learning perspective is considered to complement the TAM. The rationale for the impact of learning behaviors on continuance intention is that teachers' personal strong intentions to learn about how to apply technology in their instruction will contribute to more technology being used in their teaching [5]. Since most teachers, as digital immigrants, were born and raised before the digital age and were thus exposed to technology at a relatively older age compared to digital natives, they have more difficulties in combining technology with teaching [50]. At present, a wide range of teacher education programs are offered to better teachers' competence in applying technology in teaching, particularly for teachers in universities or in-service teachers in the workplace. While participating in technology development programs, these teachers will be more likely to engage in learning, and their learning behaviors should be beneficial for their subsequent continued technology use, on the condition that they hold a firm belief that they can better their technology competence through learning (i.e., a growth mindset). Based on the literature about students' academic engagement [51], it is expected that teachers who have a growth mindset will be positively involved in learning about how to apply technology and this will enhance their competence with more continuance intention towards technology use.

## 1.2. Literature Review

With the advancement of computer technology in the new millennium, it has become a common and indispensable tool that allows students to control their own learning to achieve a longer-term learning goal, including gaining foreign language learning experiences [52]. The purpose of the TAM is to explain the main factors of user behavior towards technology user acceptance [53]. Teachers' continuance intention is determined by different

variables. Perceived ease of use and perceived usefulness are two main components of the TAM. Perceived ease of use refers to the degree to which a person believes that using a particular system would be free of effort. Perceived usefulness refers to the degree to which a person believes that using a particular system will increase his or her job performance. Perceived ease of use and perceived usefulness affect the attitude towards using technology, which is conceptualized as an attitude towards the use of a system in the form of acceptance or rejection [53]. As a teacher becomes more accountable and interested in the use of technology in the classroom, perceptions develop, confidence takes root, interest is augmented, and concern about its use becomes commonplace. Thus, one's perceptions become an integral part of both efficacy and one's interest in having the ability to continue using technology. Furthermore, effort regulation and help seeking are two more factors. Effort regulation is the ability to monitor and sustain effort even when the content is difficult, frustrating, or boring. However, help seeking is understood as the current intention to seek help from different sources for different problems, as well as the quantity and quality of previous professional psychological helping episodes [54]. Teachers must be able to learn during and from practice since teaching knowledge is rarely fully acquired prior to or separate from practice [55]. Effort regulation and help seeking are essential here as they lead to effectiveness in instruction [56,57]. In this regard, effort regulation and help seeking can improve teachers' self-efficacy and interest, and hence, reduce their anxiety in teaching. Good teachers who succeed in using technology in instruction regulate their teaching practices and seek help from colleagues to prevent any negative emotional factors. This may help them avoid problems in using technology in their classrooms.

Moreover, the factor of facilitating conditions is considered to be a construct used in research to measure the level of perception of the user regarding the support of the organizational environment and the needed infrastructure to use the new technology. Facilitating conditions are organizational and technical infrastructure supporting the use of acquired systems in their contexts. A teacher might utilize technology resources to change some instructional behaviors in response to changing environmental conditions [5]. Hence, this factor can reduce the anxiety level of the teacher and simultaneously boost them to move back and forth between positive motives and technology implementation.

According to Dweck's theory of mindsets (2000, 2006) [58,59], individuals may hold either a growth mindset or a fixed mindset, which pertains to their beliefs about the malleability of traits such as intellect and ability. A growth mindset is characterized by the belief that these traits can be developed through effort and learning, while a fixed mindset is characterized by the belief that they are fixed and incapable of change. Research conducted by Blackwell et al. (2007) [60] suggests that individuals with a growth mindset tend to exhibit stronger learning goals and more positive beliefs about the role of effort in achieving success, and they are more inclined to engage in effort-based strategies. In education, growth-minded teachers use process-based pedagogy to foster a positive learning atmosphere, whereas fixed-minded teachers emphasize students' fundamental traits, which might lower motivation and tenacity [61]. The link between teachers' mindsets and their use of technology in the classroom has been substantiated through research, with Alshehri (2022) [31] and Teo et al. (2018) [62] both finding a significant relationship between teachers' mindsets and their use of technology in the classroom, with those having a growth mindset being more likely to utilize technology in their teaching. White (2019) [32] asserts that a "digital mindset," or confidence in one's ability to develop digital skills and adapt professional practices accordingly, is essential for individuals in the teaching profession to keep pace with the rapid development and adoption of new technologies. Teachers' mindsets can impact their use of technology and teaching practices, making it an important area of investigation. Ergen (2019) [63] also found that teachers with a growth mindset tend to have higher self-efficacy in using technology as they believe in their ability to acquire and apply new skills, while those with a fixed mindset may have lower self-efficacy and less willingness to integrate technology in their teaching.

Another important factor is self-efficacy. This is the capacity to believe in one's own ability to accomplish tasks [64]. Teacher self-efficacy, specifically, refers to a teacher's belief in their ability to effectively carry out their teaching responsibilities within a particular setting [65]. High self-efficacy increases a teacher's likelihood of being prepared, enthusiastic, and resilient in the face of challenges. The use of technology in the classroom can impact teacher self-efficacy, with those who possess self-efficacy in technology use being more motivated to use it in teaching [66]. However, teachers often have low levels of competence and self-efficacy with technology [67,68]. Factors that may impact teacher self-efficacy with technology include age and gender [69], computer experience [70], and school support [71]. In terms of basic and advanced computer skills, for instance, Scherer and Siddiq (2015) [69] found that male teachers tended to have higher levels of self-efficacy. These findings highlight the complexity of the relationship between self-efficacy and technology use in the classroom. For example, high self-efficacy and interest in technology can lead to a greater intention to continue using technology in the future, while low self-efficacy can result in a decreased likelihood of technology use [63]. These interrelated factors illustrate the importance of considering self-efficacy in discussions about teacher technology use.

Professional development that sparks teachers' interest and motivates them to engage in exploration and learning can enhance their ability to adapt to new demands and challenges in their profession and improve their students' learning outcomes [72]. Interest is a mental and emotional state marked by rapt attention, pleasure, and excitement that develop in response to exposure to intriguing objects or ideas [73]. It is accompanied by the maturing of a person's affective and cognitive dimensions and may stimulate an intrinsic desire to engage in a certain activity or topic [74]. Hidi (2006) [74] suggests that teachers may have the greatest influence on their students' achievement if they assist them to develop academically relevant interests. An adequate level of interest in using technology is crucial for teachers to effectively integrate it into their teaching practices [33]. However, maintaining this level of interest requires ongoing professional development and support from school leaders. Online teaching, for example, has become a popular choice among EFL teachers and students due to its ability to foster learner autonomy and identity formation through hybrid uses of language [75]. Despite English teachers in China having positive attitudes towards technology use, their actual use of technology in teaching is limited and peripheral [76]. Therefore, promoting professional development that sparks teachers' interest and motivates them to engage in exploration and learning to improve their students' learning outcomes is of high importance.

Technology has gained attention in education for its potential to improve teaching and learning outcomes [77]. Continuance intention, or the intention to continue using a technology after initial acceptance, has been studied in relation to ICT adoption and usage [78,79]. Motivational beliefs such as self-efficacy and interest, learning behaviors, and facilitating conditions can influence continuance intention [43,80]. Technology self-efficacy, a supportive atmosphere, interest, and a growth mindset all had favorable impacts on the intentions of teachers of English as a second language to keep using technology in their classrooms [43,80]. For instance, a school culture that values and encourages the use of technology in education can motivate ESL teachers to persist in incorporating technology into their teaching methods. In contrast, anxiety had a negative impact on teachers' technology continuance intention. The objectives that should drive professional development programs for teachers include fostering a growth mindset and encouraging instructors to seek help when needed.

*1.3. Research Aims and Hypotheses*

Based on the review of previous studies, this research aimed to examine the effects of help seeking, interest, effort regulation, growth mindset, facilitating conditions, perceived usefulness, and perceived ease of use on continuance intention, and the mediating role of self-efficacy in the relationship between the above-mentioned factors and continuance inten-

tion. The following hypotheses were tested during the course of this investigation. Figure 1 illustrates the relationship between the variables and hypotheses of the present study:

➢ There are direct effects of effort regulation ($H_1$), facilitating conditions ($H_2$), interest ($H_3$), growth mindset ($H_4$), help seeking ($H_5$), perceived ease of use ($H_6$), and perceived usefulness ($H_7$) on teachers' continuance intention to use technology in EFL instruction;

➢ There are indirect effects of effort regulation ($H_8$), facilitating conditions ($H_9$), interest ($H_{10}$), growth mindset ($H_{11}$), help seeking ($H_{12}$), perceived ease of use ($H_{13}$), and perceived usefulness ($H_{14}$) on teachers' continuance intention to use technology in EFL instruction through self-efficacy.

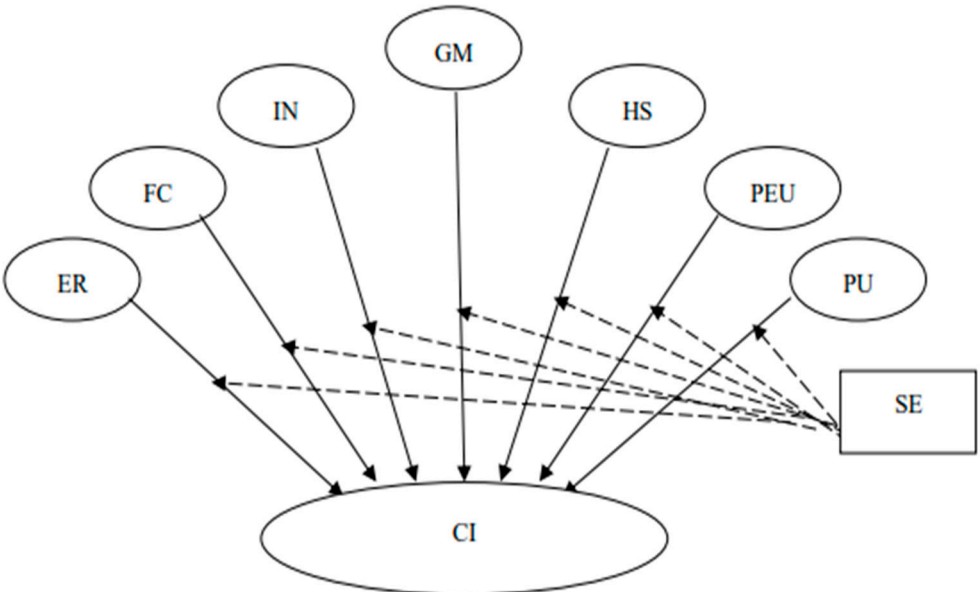

**Figure 1.** The hypothesized study model.

## 2. Methodology

### 2.1. Research Design

This study focused on teachers' continuance intention to use technology. The TAM was used. This study is quantitative. A cross-sectional research design was adopted as it helps in the collection of data from a wide range of participants and, hence, to explore and measure the interactions that exist between variables. The authors are a group of international researchers from China and other two countries, Algeria and Myanmar. Their identities range widely from professors, postdoctoral fellows, to Ph.D. students. They are composed of both female and male researchers. None of them had any effect on the choice of the scope and aims of the study, nor did they have any effect on the research methods adopted to accomplish the study. To provide a relevant and trustworthy example of technology-based teaching, which may influence the education systems of other nations, they all felt the need to investigate students' continuance intention to use technology in a highly developed country, namely China, that strongly supports the implementation of technology in education. To obtain opinions from a large number of Chinese participants, they opted for a quantitative strategy based mostly on a cross-sectional research design.

### 2.2. Procedures

The researchers were aware that, for any study, the validity and reliability of the questionnaire is a significant issue to assure the correctness and validity of the results. To attain this, following ethical guidelines in data collection is critical and indispensable. In this regard, a written agreement was obtained from the Research Ethics Committee of the School of Foreign Languages, Huazhong University of Science and Technology

(HUST), to collect data for study aims. English teachers were selected using a purposive sampling procedure from different junior secondary schools in Western China. Purposive sampling entails picking units based on certain criteria that are necessary and appropriate for the study [5,6]. The selection process was based on the location of the participants. The researchers sought to achieve balance in the numbers of teachers from rural and urban schools. The reason for this is the fact that teaching in rural and urban areas in the western parts of China is not the same. There are some differences in terms of the supply and use of technology, teaching quality, academic achievements, the support provided to schools and teachers, contextual conditions inside the schools, and the level of the students. The researchers started by contacting the junior secondary school principals. The first researcher explained the study to the principals and asked them to contact their teachers. Meetings with teachers were arranged with the help of the principals. However, most of the teachers were contacted online due to the effects of the COVID-19 pandemic. The first researcher explained the study to the teachers and confirmed to them that the data would be kept private and used only for the study aims. All of them agreed to participate. Then, a consent letter was obtained from each of the participants.

A total of 601 teachers from junior secondary schools in rural and urban areas of Western China participated in the current study. Among these 601 teachers, there were some who had been contacted at their schools by some friends of the researchers, whereas others were contacted online using WeChat and QQ applications. The participants who were contacted online could not be reached face-to-face due to the COVID-19 pandemic situation in their cities. In this regard, only fully answered questionnaire forms were taken into account. In the end, a total of 459 English teachers from 15 junior secondary schools in Western China were counted since they provided complete answers to the questionnaire.

*2.3. Participants*

The final sample of participants comprised 459 English language teachers from different regions in Western China, including Guangxi, Guizhou, Gansu, Qinghai, Xinjiang, Yunnan, Ningxia, Shaanxi, and Enshi Autonomous Prefecture in Hubei, during the 2021–2022 academic year. Sixteen junior secondary schools participated in this study. Most of these schools were based in rural and urban areas in Western China and they all used technology in instruction and learning. There were eight located in urban areas, whereas eight were located in rural areas in Western China. As shown in Table 1, of the 459 teachers, 216 were male and 243 were female; 206 teachers had a bachelor's degree, 163 teachers had a master's degree, whereas 90 teachers had other degrees. The ages of the teachers were categorized in five categories: 69 teachers were 30 years old or younger, 133 teachers were between 31 and 35 years old, 82 teachers were between 36 and 40 years old, 105 teachers were between 41 and 45 years old, and 70 teachers were 46 or above.

**Table 1.** Participants' profiles.

| Demographic Variables | Frequency | Percentage | M | SD |
|:---:|:---:|:---:|:---:|:---:|
| Gender | 459 | 100 | | |
| Male | 216 | 47.1 | 1.529 | 0.499 |
| Female | 243 | 52.9 | | |
| Education Level | 459 | 100 | | |
| Bachelor | 206 | 44.9 | | |
| Master | 163 | 35.5 | 1.747 | 0.763 |
| Others | 90 | 19.6 | | |
| Age | 459 | 100 | | |
| 30 years old and less | 69 | 15 | | |
| 31–35 | 133 | 29 | | |
| 36–40 | 82 | 17.9 | 2.943 | 1.315 |
| 41–45 | 105 | 22.9 | | |
| 46 years old and above | 70 | 15.3 | | |

*2.4. Research Instrument*

The questionnaire of Bai et al. (2021) [7] was used in the current study. The tool was initially derived from Pintrich et al. (1991) [81], Morris et al. (2003) [82], Dweck (2006) [59], Chiu and Wang (2008) [83], and Liaw and Huang (2013) [80], and was modified to fit the context of English teaching. As shown in Table 2, the scale entails thirty-seven items distributed among nine factors: facilitating conditions (four items), self-efficacy (four items), interest (four items), perceived ease of use (four items), perceived usefulness (four items), growth mindset (four items), effort regulation (five items), help seeking (four items), and continuance intention (four items).

**Table 2.** Questionnaire factors and items.

| Scale Factors | Number of Items |
|---|---|
| Facilitating conditions | 4 |
| Self-efficacy | 4 |
| Interest | 4 |
| Perceived ease of use | 4 |
| Perceived usefulness | 4 |
| Growth mindset | 4 |
| Effort regulation | 5 |
| Help seeking | 4 |
| Continuance intention | 4 |

This scale had been used by Bai et al. (2021) [7] in a Chinese context, in Hong Kong, but to guarantee its suitability for the Western China context, it was provided to some experts for evaluation. The five experts suggested keeping the scale as it had been formulated by Bai et al. (2021) [7] since the items are clear and can fit also the Western China educational context. However, they suggested conducting pilot testing of the scale, to confirm its validity and reliability. The pilot testing was then performed with 106 teachers to support the validity evaluation of the experts. The Cronbach's alpha ($\alpha$) was found to be adequate, with a value of 0.81; hence, the validity was determined to be 0.91.

## 3. Data Analysis

The analysis went through exploratory factor analysis and confirmatory factor analysis. Data entry was carried out using Statistical Product and Service Solutions (SPSS) Software (Version 22.0) and SmartPLS. As the data entry was one of the essential processes in this study, it was carried out with particular attention to obtain valid results. Finally, data analysis and interpretation were conducted.

## 4. Research Instrument Validity and Reliability

This study aimed to examine the effects of growth mindset, interest, facilitating conditions, effort regulation, help seeking, perceived ease of use, and perceived usefulness on teachers' continuance intention to use technology in teaching, and the mediation effect of self-efficacy on these relationships. The factorial validity was examined to confirm the validity of the scale. The KMO obtained in this study (KMO = 0.951) was greater than the values suggested by previous studies [84]. BST was found to be significant ($x^2$ = 11,914.1892; $p \leq 0.001$). Hence, normal distribution of data with multiple variables was affirmed. These results demonstrated that the questionnaire was appropriate for factor analysis [85,86]. The most likely number of variables to match the data was nine. As indicated in Table 3 and Figure 2, the number of factors that best suited the data was most likely nine. The initial EFA for 37 items with eigenvalues revealed a nine-factor structure, which was greater than 1 that could be extracted, accounting for 73.106% of the total variance [84,86].

**Table 3.** Eigenvalues and percentage of variance explaining the factors.

| Component | Initial Eigenvalues | | | Extraction Sums of Squared Loadings | | | Rotation Sums of Squared Loadings | | |
|---|---|---|---|---|---|---|---|---|---|
| | Total | % of Variance | Cumulative % | Total | % of Variance | Cumulative % | Total | % of Variance | Cumulative % |
| ER | 15.100 | 40.810 | 40.810 | 15.100 | 40.810 | 40.810 | 7.029 | 18.996 | 18.996 |
| FC | 2.781 | 7.516 | 48.327 | 2.781 | 7.516 | 48.327 | 4.054 | 10.956 | 29.952 |
| IN | 2.078 | 5.616 | 53.942 | 2.078 | 5.616 | 53.942 | 3.326 | 8.988 | 38.941 |
| GM | 2.009 | 5.430 | 59.373 | 2.009 | 5.430 | 59.373 | 2.796 | 7.558 | 46.498 |
| GS | 1.219 | 3.295 | 62.668 | 1.219 | 3.295 | 62.668 | 2.437 | 6.588 | 53.086 |
| HS | 1.118 | 3.022 | 65.690 | 1.118 | 3.022 | 65.690 | 2.432 | 6.573 | 59.659 |
| PEU | 1.070 | 2.731 | 68.421 | 1.070 | 2.731 | 68.421 | 2.409 | 6.511 | 66.170 |
| PU | 1.014 | 2.417 | 70.837 | 1.014 | 2.417 | 70.837 | 1.540 | 4.162 | 70.332 |
| ER | 1.001 | 2.268 | 73.106 | 1.001 | 2.268 | 73.106 | 1.026 | 2.774 | 73.106 |

Extraction Method: Principal Component Analysis.

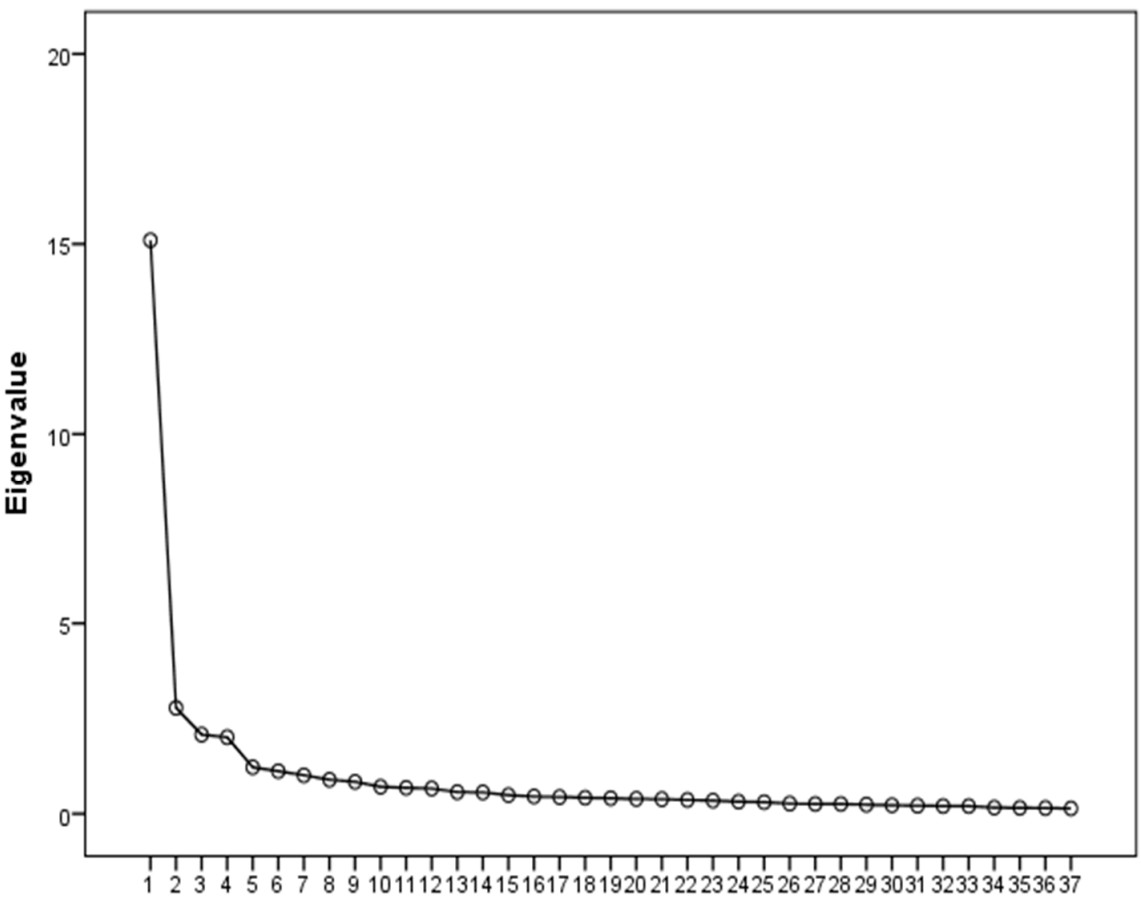

**Figure 2.** Scree plot of the study questionnaire.

The initial EFA with eigenvalues for 37 items revealed a nine-factor structure. The results showed that the same 37 items that were distributed among nine factors as their factor loads were all higher than 0.40: facilitating conditions (four items) with a factor load range between 0.730 and 0.488, self-efficacy (four items) with a factor load range between 0.849 and 0.578, interest (four items) with a factor load range between 0.814 and 0.727, perceived ease of use (four items) with a factor load range between 0.901 and 0.884, perceived usefulness (four items) with a factor load range between 0.808 and 0.780, growth mindset (four items) with a factor load range between 0.674 and 0.606, effort regulation (five items) with a factor load range between 0.723 and 0.643, help seeking (four items) with a factor load range between 0.643 and 0.430, and continuance intention (four items) with a factor load range between 0.771 and 0.425. Confirmatory factor analysis was also executed

to confirm the research instrument items and all loading values were higher than 0.48, and all factor loadings were statistically significant at $p < 0.01$.

In addition, the measurement model was assessed using multiple fit indices, including $x^2/DF = 2.498$, the root mean square error of approximation (RMSEA) = 0.067, the comparative fit index (CFI) = 0.921, the goodness of fit index (GFI) = 0.966., and the Tucker–Lewis index (TLI) = 0.903. All these values of the fit indices seemed to be appropriate [84] and they confirmed the validity of the proposed model, and that the final nine-factor model fit well.

The Cronbach's alpha ($\alpha$) values for each component were 0.783, 0.711, 0.862, 0.793, 0.792, 0.834, 0.773, 0.709 and 0.807, respectively. All of these values were suitable and acceptable for this measurement [87]. The composite reliability (CR) values for each factor were 0.765, 0.834, 0.785, and 0.841. All the average variance extracted (AVE) values were higher than 0.50, indicating a good approximation of validity [86,88]. In order to evaluate discriminant validity, each factor that contained the AVE was also tested with the squared correlation. The proof of discriminant validity was satisfactory [84,89].

## 5. Results

SmartPLS was employed to test the model's explanatory capacity based on adopting resampling methods to simplify calculating the PLS coefficient's significance [90]. The fit indices were assessed and proved the appropriateness and validity of the model since the following values were found to be suitable and high [84]: SRMR = 0.057, d_ULS = 2.282, d_G = 0.814, Chi-Square ($x^2$) = 2181.486, NFI = 0.903, and rms Theta = 0.122. Table 4 presents the construct reliability and validity, whereas Table 5 presents discriminant validity. All values were adequate and confirmed the model quality [91].

**Table 4.** Construct reliability and validity of the study model.

| Variables | $\alpha$ | rho_A | CR | AVE |
|:---:|:---:|:---:|:---:|:---:|
| CI | 0.809 | 0.891 | 0.879 | 0.661 |
| ER | 0.832 | 0.854 | 0.881 | 0.599 |
| FC | 0.809 | 0.811 | 0.877 | 0.724 |
| IN | 0.900 | 0.901 | 0.938 | 0.833 |
| GM | 0.820 | 0.822 | 0.893 | 0.735 |
| GS | 0.898 | 0.900 | 0.929 | 0.766 |
| HS | 0.750 | 0.811 | 0.780 | 0.527 |
| PEU | 0.898 | 0.899 | 0.929 | 0.767 |
| PU | 0.862 | 0.869 | 0.907 | 0.709 |

Note: $\alpha$ = Cronbach's alpha, CR = composite reliability, AVE = average variance extracted.

**Table 5.** Discriminant validity of the study model.

| | CI | ER | FC | GI | GM | GS | HS | PEU | PU |
|:---:|:---:|:---:|:---:|:---:|:---:|:---:|:---:|:---:|:---:|
| CI | 0.831 | | | | | | | | |
| ER | 0.731 | 0.774 | | | | | | | |
| FC | 0.462 | 0.385 | 0.851 | | | | | | |
| IN | 0.643 | 0.642 | 0.396 | 0.931 | | | | | |
| GM | 0.706 | 0.742 | 0.367 | 0.694 | 0.857 | | | | |
| GS | 0.553 | 0.511 | 0.454 | 0.558 | 0.556 | 0.875 | | | |
| HS | 0.684 | 0.662 | 0.371 | 0.523 | 0.597 | 0.450 | 0.726 | | |
| PEU | 0.480 | 0.577 | 0.245 | 0.609 | 0.634 | 0.554 | 0.448 | 0.876 | |
| PU | 0.710 | 0.756 | 0.391 | 0.744 | 0.759 | 0.523 | 0.581 | 0.614 | 0.842 |

Then, the hypotheses were assessed as presented in Table 6 below. Concerning the assessment of the direct effect, the bootstrap resampling method with 5000 resamples [92] was carried out. The direct effects of all hypotheses were accepted, except the effect of interest on continuance intention (β = 0.068, Std = 0.053, t = 1.270, *p*-value = 0.205). Table 6 shows the significant positive effects of effort regulation on continuance intention (β = 0.217, Std = 0.060, t = 3.637, *p*-value = 0.000), of facilitating conditions on continuance intention (β = 0.086, Std = 0.031, t = 2.754, *p*-value = 0.006), of growth mindset on continuance intention (β = 0.179, Std = 0.053, t= 3.346, *p*-value = 0.001), of help seeking on continuance intention (β = 0.259, Std = 0.049, t = 5.251, *p*-value = 0.000), of perceived ease of use on continuance intention (β = 0.104, Std = 0.040, t = 2.577, *p*-value = 0.010), and of perceived usefulness on continuance intention (β = 0.160, Std = 0.060, t = 2.645, *p*-value = 0.008).

**Table 6.** Direct effects among the study variables.

| Hypotheses | Direct Effect | β | M | SD | *t*-Test Value | *p*-Value | Decision |
|---|---|---|---|---|---|---|---|
| H₁ | ER→CI | 0.217 | 0.229 | 0.060 | 3.637 | 0.000 | Supported |
| H₂ | FC→CI | 0.086 | 0.086 | 0.031 | 2.754 | 0.006 | Supported |
| H₃ | IN→CI | 0.068 | 0.062 | 0.053 | 1.270 | 0.205 | Unsupported |
| H₄ | GM→CI | 0.179 | 0.171 | 0.053 | 3.346 | 0.001 | Supported |
| H₅ | HS→CI | 0.259 | 0.262 | 0.049 | 5.251 | 0.000 | Supported |
| H₆ | PEU→CI | 0.104 | 0.106 | 0.040 | 2.577 | 0.010 | Supported |
| H₇ | PU→CI | 0.160 | 0.160 | 0.060 | 2.645 | 0.008 | Supported |

To test the mediation of self-efficacy, Preacher and Hayes's (2008) [93] method was used, and *p*-values of indirect effects were obtained through bootstrapping with 5000 resamples [92]. The results confirmed significant indirect effects on continuance intention through the mediation of self-efficacy for only two variables, facilitating conditions (β = 0.027, Std = 0.012, t = 2.194, *p*-value = 0.029) and perceived ease of use (β = 0.030, Std = 0.014, t = 2.192, *p*-value = 0.029); hence, the two related hypotheses were supported. However, the analyses revealed that self-efficacy had no mediation effect on the effects of effort regulation on continuance intention (β = 0.003, Std = 0.009, t = 0.329, *p*-value = 0.742), of interest on continuance intention (β = 0.018, Std = 0.011, t = 1.651, p-value = 0.099), of growth mindset on continuance intention (β = 0.015, Std = 0.011, t = 1.352, *p*-value = 0.177), of help seeking on continuance intention (β = 0.007, Std = 0.007, t = 1.068, *p*-value = 0.286), or of perceived usefulness on continuance intention (β = 0.003, Std = 0.009, t = 0.397, *p*-value = 0.691); therefore, the related hypotheses were unsupported. The results are illustrated in Table 7 and Figure 3.

**Table 7.** Indirect effects among the study variables.

| Hypotheses | Direct Effect | β | M | SD | *t*-Test Values | *p*-Values | Decision |
|---|---|---|---|---|---|---|---|
| H₈ | ER→SE→CI | 0.003 | 0.003 | 0.009 | 0.329 | 0.742 | Unsupported |
| H₉ | FC→SE→CI | 0.027 | 0.028 | 0.012 | 2.194 | 0.029 | Supported |
| H₁₀ | IN→SE→CI | 0.018 | 0.019 | 0.011 | 1.651 | 0.099 | Unsupported |
| H₁₁ | GM→SE→CI | 0.015 | 0.015 | 0.011 | 1.352 | 0.177 | Unsupported |
| H₁₂ | HS→SE→CI | 0.007 | 0.008 | 0.007 | 1.068 | 0.286 | Unsupported |
| H₁₃ | PEU→SE→CI | 0.030 | 0.031 | 0.014 | 2.192 | 0.029 | Supported |
| H₁₄ | PU→SE→CI | −0.003 | −0.003 | 0.009 | 0.397 | 0.691 | Unsupported |

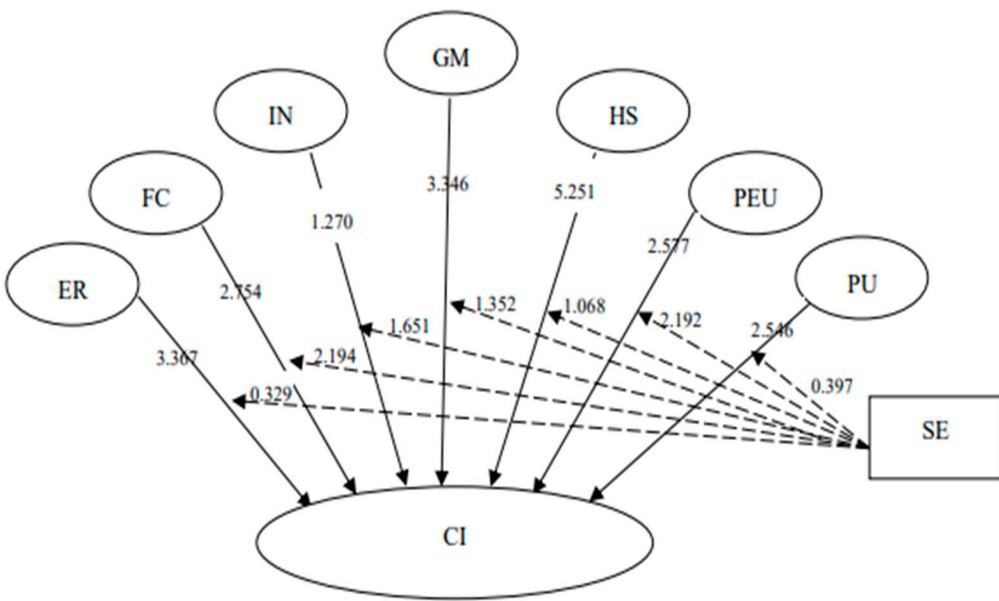

**Figure 3.** Structural model.

## 6. Discussion

The current study aimed to investigate the continuance intention to use technology in teaching among Chinese teachers of English in junior secondary schools in Western China. After following various procedures to obtain a large number of participants, as presented above, the final sample included 459 teachers from Guangxi, Guizhou, Gansu, Qinghai, Xinjiang, Yunnan, Ningxia, Shaanxi, and Enshi Autonomous Prefecture in Hubei. After checking the surface validity, pilot testing, final distribution of the questionnaire, and lastly, measuring the questionnaire's validity and reliability, the questionnaire validity and reliability were confirmed to entail thirty-seven items.

The hypotheses of the present study were of two types, including direct and indirect effects. The direct effect hypotheses included the effects of the growth mindset, facilitating conditions, interest, effort regulation, help seeking, perceived ease of use, and perceived usefulness on the continuance intention to use technology in teaching. The results revealed that a direct effect existed between growth mindset, facilitating conditions, effort regulation, help seeking, perceived ease of use, and perceived usefulness and the continuance intention to use technology in teaching, since the P values were less than 0.05. These results were similar to the study of Tang et al. (2021) [26], which showed that the growth mindset, help seeking, and perceived usefulness significantly determined teachers' intention to adopt mobile technology as an enhanced teaching platform. Meanwhile, Tang et al. (2021) [26] found that the perceived ease of use factor had no effect. On the other hand, it has been shown that the perceived ease of use and perceived usefulness of mobile technology have a direct influence on teachers' intentions to use technology in their classrooms [34,35]. When tutors have greater expertise or comfort with the technology, they will find it simpler to employ mobile devices to aid them in online instruction [26,34]. This, in turn, will influence their beliefs and actions about the continuing adoption of the technology [94]. The result related to the effect of effort regulation on continuance intention contradicted the study of Bai et al. (2021) [7], which found that effort regulation did not have a positive direct effect on continuance intention among Chinese primary teachers. However, the study of Bai et al. (2021) [7] was conducted in Hong Kong and the authors explained that the primary school teachers there did not need much effort to acquire knowledge about how to use technology tools, while the participants in the current study were teachers in junior secondary schools in Western China and they needed to become familiar with using technology, especially the technology that has developed and spread recently in China and the new inventions being introduced in education. Some studies that have

been conducted on the topic of people's inclination to accept and use different educational technologies, such as e-learning and mobile learning, have placed an emphasis on the role that facilitating conditions play in this phenomenon [7,25]. These studies highlight how crucial it is to create settings that are pleasant for users in order to encourage broad adoption of technologies [95]. However, our analysis revealed that no effect exists between interest and continuance intention. This result contradicted the study of Bai et al. (2021) [7], which found significant direct and positive effects of interest on continuance intention. This finding also differed from the study of Lai and Chen (2011) [96], which found that teachers' satisfaction in using technology was directly and positively connected with their adoption of blogs. The possible reason for this is that, as the use of technology has become a regular and normal issue in teaching, the intention to continue using technology has become normal but it still demands effort, positive perceptions, cooperation between teachers, and a conducive environment.

Concerning the indirect effects that were explored between variables in which self-efficacy was a mediator variable, varying results were obtained. Surprisingly, indirect effects of growth mindset, interest, effort regulation, help seeking, and perceived usefulness on continuance intention through self-efficacy were not confirmed. The explanation could be that self-efficacy may have been at a low level among the participants [67,68], which made its contribution as a mediator without having an effect, while effort, positive perceptions, cooperation between teachers, and their interests had a considerable influence on continuance intention, as confirmed above. Therefore, it is necessary to make appropriate plans to promote self-efficacy among Chinese teachers of English language in junior secondary schools. Meanwhile, the results confirmed the indirect effects of facilitating conditions and perceived ease of use on continuance intention through self-efficacy. These results were similar to the study of An et al. (2022) [97], which found that technological self-efficacy mediated the relationship between continuance intention to use technology and self-directed learning. This result also supported the study of Sharma and Saini (2022) [37], which revealed that self-efficacy plays a moderating role in the relationship between continuance intention and the actual use of technology. It is reasonable to assume that when the surrounding environment is favorable for instructors and they view a technology to be very user-friendly, their positive beliefs about incorporating technology in teaching will grow, hence increasing their utilization of technology.

## 7. Conclusions

There has never been a more crucial moment for English instructors to become adept with and manage technology-based teaching. Therefore, it is essential to understand the factors driving the continuance intention of instructors to use technology in the classroom. This research aimed to uncover the factors that affect English language teachers' continuance intention to use technology in the junior secondary schools of Western China. It investigated the relationships between growth mindset, facilitating conditions, effort regulation, help seeking, interest, perceived ease of use, and perceived usefulness and continuance intention, as well as the role of self-efficacy as a mediator. Teachers' intention to continue using technology was shown to be influenced by the growth mindset, facilitating conditions, effort regulation, help seeking, perceived ease of use, and perceived usefulness. However, the research demonstrated that interest had no role or effect. Facilitating conditions and perceived ease of use did have an indirect effect on continuance intention through self-efficacy. However, self-efficacy did not mediate the effects of the growth mindset, interest, effort regulation, help seeking, and perceived usefulness on continuance intention. As a result, it is recommended that instructors attempt to increase their self-efficacy in order to increase students' motivation to persevere with technology. These findings suggest that policies and practices for EFL-instruction-based technology over the next few years can take a variety of forms. The government should prioritize the planned improvement of technology-based instruction training programs in order to increase teachers' interest, motivation, and perspectives, thereby ensuring their intention to

continue utilizing technology. In addition, the government should increase the supply of necessary technology devices for teachers while indirectly enhancing teachers' intention to use technology. Similarly, teachers should participate in training programs offered by the government and other agencies and collaborate with government, colleagues, school principals, and researchers to improve the efficacy of the technology used in instruction, which may lead to an increase in the proportion of tutors who intend to continue using technology.

## 8. Limitations and Future Research Directions

There were several limitations to this research. Age, gender, and other socio-demographic factors were disregarded. For the sake of precision, future studies might benefit from the incorporation of demographic data when estimating how likely participants are to maintain their current levels of continuance intention to use technology. This study focused on the effects of the growth mindset, facilitating conditions, interest, effort regulation, help seeking, perceived ease of use, and perceived usefulness on continuance intention to use technology in teaching. These factors may not provide an accurate or complete depiction of what influences continuance intention. Future research could examine a broad variety of past studies, or conduct qualitative investigations, to explore additional factors. This research also focused on continuance intention, which may not provide a complete picture of the actual use of technology in education. Future studies could address actual technology use. A total of 459 junior high school instructors of English in Western China responded to the survey, but this did not capture the demographics of professors or the variety of students at other institutions. Consequently, care must be used when extrapolating these results to the whole teaching force. The recruitment of educators in a wide variety of institutions is strongly recommended for future studies for the reason that junior high school English instructors were the primary focus of this research. Other areas of China (the east, the south, and the north), educational levels (including higher education institutions), and fields of study (hard sciences, social sciences, and the humanities) might be the focus of future studies. The employment of a single research approach throughout the survey also did not add to the study's credibility. There are different opinions as to whether or not it could fully and accurately represent the opinions of the people who participated. Future studies on this topic would be more convincing if they included data on interviews and classroom observations.

**Author Contributions:** Conceptualization, Y.X., A.B. and Y.L.; Methodology, A.H.A.-Q.; Software, Y.M.A.; Validation, J.X. and A.H.A.-Q.; Formal analysis, A.H.A.-Q. and Y.M.A.; Investigation, Y.X. and J.X.; Resources, A.B. and A.K.; Data curation, A.B. and J.X.; Writing—original draft, A.B. and A.K.; Writing—review & editing, Y.X., A.B. and J.X.; Supervision, A.B.; Project administration, A.B. and Y.L.; Funding acquisition, Y.X. All authors have read and agreed to the published version of the manuscript.

**Funding:** This research received no external funding.

**Institutional Review Board Statement:** This study was conducted in accordance with all the required ethical considerations and practices. The study was approved by the Research Ethics Committee of the School of Foreign Languages, Huazhong University of Science and Technology (HUST).

**Informed Consent Statement:** Informed consent was gathered from all participating teachers. Confidentiality was maintained by not requesting names or any other information that would identify the teachers involved. The subjects were informed of their right to withdraw from the investigation at any time.

**Data Availability Statement:** Data will be made available on request.

**Conflicts of Interest:** No potential conflict of interest was reported by the authors.

## Abbreviations

EFL: English as a foreign language, HS: help seeking, IN: interest, ER: effort regulation, GM: growth mindset, FC: facilitating conditions, PU: perceived usefulness, PEU: perceived ease of use, SE: self-efficacy, CI: continuance intention.

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
