# Peer review of "A Study on Teachers’ Continuance Intention to Use Technology in English Instruction in Western China Junior Secondary Schools"

_sustainability, doi:10.3390/su15054307_

Round 1

Reviewer 1 Report

Please provide a statement of third-party approval that you secured to conduct this study (e.g., Institutional Review Board for the Protection of Human Subjects) or if your local context does not require such oversight, then please indicate this and describe how you ensured ethical research practice to protect participants’ safety, privacy, and confidentiality. If the study was deemed to be exempted or excluded from IRB review, please make note of it. 

Please add an about the researcher section so you tell us more about you as the researcher(s) and your connection to this study. How does this align with personal interests, professional work, etc., to help the reader place you directly in the center of your work?

Include a greater discussion of how your participants were identified and recruited. Also, include information about the schools were participants worked. 

Discuss further the implications your research has for pertinent stakeholders (e.g., practice suggestions for practitioners, or policy considerations for administrators) and make sure comments are directly connected with the study you conducted. 

Author Response

Manuscript title:

A Study on Teachers’ Continuance Intention to Use Technology in English Instruction in Western China Junior Secondary Schools

Dear Reviewer,

Sub. Submission of Revised Paper " A Study on Teachers’ Continuance Intention to Use Technology in English Instruction in Western China Junior Secondary Schools"

   Article ID: sustainability-2192501

  Thank you for your email dated 30/01/2023 enclosing the reviewers' comments. We have carefully reviewed the comments and have revised the manuscript accordingly. the paper was edited by MDPI editing certificate. The responses are given in a point-by-point manner below.

S/N

Reviewer 1’s comments

Corrections by authors

1.        

  Please provide a statement of third-party approval that you secured to conduct this study (e.g., Institutional Review Board for the Protection of Human Subjects) or if your local context does not require such oversight, then please indicate this and describe how you ensured ethical research practice to protect participants’ safety, privacy, and confidentiality. If the study was deemed to be exempted or excluded from IRB review, please make note of it. 

This statement was added before references section:

Ethical Approval:  This study was conducted in accordance with all the required ethical considerations and practices. The study was approved by the Research Ethics Committee of the School of Foreign Languages, Huazhong University of Science and Technology (HUST).

In addition, this expression was added in procedure section:

In this regard,  written agreement was obtained from the Research Ethics Committee of the School of Foreign Languages, Huazhong University of Science and Technology (HUST),

2.        

Please add an about the researcher section so you tell us more about you as the researcher(s) and your connection to this study. How does this align with personal interests, professional work, etc., to help the reader place you directly in the center of your work?

 Authors information were added:

Yi Xie, is a PhD student of School of Foreign Languages, Huazhong University of Science and Technology (HUST) in Wuhan, with a research direction of second language acquisition, foreign language education and teacher professional development.

Azzeddine Boudouaia  is a Postdoctoral Fellow in Artificial Intelligence in education at the College of Education, Zhejiang University, China. He has a Ph.D. in curriculum and teaching methodology. His research interests include curriculum studies, technology and artificial intelligence in EFL education, EFL teaching approaches, and teacher professional development.

Abdo Hasan AL-Qadri is a faculty member at Xi’an Eurasia University, China.  His research interests include educational and psychological measurements and evaluation, educational psychology, different educational sciences areas.

Asma Khattala is pursuing a Ph.D. degree in Teaching English as a Foreign Language (TEFL) with a concentration on cultivating EFL students' cross-cultural awareness at Sétif 2 University, Algeria. Her research interests cover syllabus design, intercultural communication instruction in the EFL context, teacher development, and empowerment, and the impact of ICT and AI on EFL instruction.

Yan Li is a professor working in College of Education, Zhejiang University, China. She is the Director of Research Center for AI in Education and Director of Department of Curriculum and Learning Sciences, Zhejiang University. Her research interests include distance education, ICT education, AI in education, diffusion of educational innovations, etc.

Ya Min Aung is an associate professor at the department of teacher education, Mynmar. Her research interests focus on comparative education, teacher education, instruction, and learning.

Jinfen Xu is professor in Huazhong University of Science and Technology (HUST) and a doctoral supervisor and post-doctoral cooperative supervisor in foreign language education, second language acquisition and teacher development.

3.        

Include a greater discussion of how your participants were identified and recruited. Also, include information about the schools were participants worked.

These statements were added:

The selection process was based on the location of participants. The researchers affirmed to gain equilibrium in the number of teachers from rural and urban schools. The reason lays in the fact that teaching in rural and urban areas in western part of China is not the same. There are some differences in terms of the supplication of and use of technology and teaching quality, academic achievements, support provided to schools and teachers, contextual conditions inside the schools, and the level of students.

These statements about schools were also added in participants section:

There were sixteen junior secondary schools participated in this study. Most of these schools are based in rural and urban areas in Western China and they all use technology in instruction and learning. There were eight located in urban areas, whereas eight located in rural areas in Western China

4.        

 Discuss further the implications your research has for pertinent stakeholders (e.g., practice suggestions for practitioners, or policy considerations for administrators) and make sure comments are directly connected with the study you conducted.

These implications were added in conclusion section:

These findings suggest that policy and practice in EFL instruction-based technology over the next several years can take a variety of approaches. The government should prioritize the planned improvement of technology-based instruction training programs in order to increase teachers' interests, motivation, and perspectives, thereby ensuring their intention to continue utilizing technology. In addition, the government should increase the supply of necessary technology devices for teachers while indirectly enhancing teachers' intention to use technology. Similarly, teachers should participate in training programs offered by the government and other agencies and collaborate with government, colleagues, school's principals, and researchers to improve the efficacy of technology in instruction, which may lead to an increase in the proportion of tutors who intend to continue using technology.

Reviewer 2 Report

This is a study that addresses important research questions. The authors are advise to consult methodological sources that discuss structural equation models and causality. This method does not aim at the "direct effect." A good source, available in full text: https://ftp.cs.ucla.edu/pub/stat_ser/r393-reprint.pdf

Or, if this is a causal analysis, additional explanations of latent causality, inferences, and the effect of cofounding variables are needed.

Author Response

Manuscript title:

A Study on Teachers’ Continuance Intention to Use Technology in English Instruction in Western China Junior Secondary Schools

Dear Editor(s) and Reviewers,

Sub. Submission of Revised Paper " A Study on Teachers’ Continuance Intention to Use Technology in English Instruction in Western China Junior Secondary Schools"

   Article ID: sustainability-2192501

  Thank you for your email dated 30/01/2023 enclosing the reviewers' comments. We have carefully reviewed the comments and have revised the manuscript accordingly. The responses are given in a point-by-point manner below. 

S/N

Reviewer 2’s comments

Corrections by authors

1.        

This is a study that addresses important research questions. The authors are advise to consult methodological sources that discuss structural equation models and causality. This method does not aim at the "direct effect." A good source, available in full text: https://ftp.cs.ucla.edu/pub/stat_ser/r393-reprint.pdf

Or, if this is a causal analysis, additional explanations of latent causality, inferences, and the effect of cofounding variables are needed.

It has been added in the main manuscript

Reviewer 3 Report

the paper offers the results of an interesting research tackling  important issues. However the way it is presented makes the reading difficult and should require some further efforts to better elaborate notions facts and results. In particular I think that the use of a large number of acronyms is confusing and results in a sort of treasure hunt of meanings. In some  cases words could be reported extensively rather than using acronyms and in any case a table of acronym disambiguation should be presented in order to allow the reader to have a quick reference when needed.

the literature review can be improved, further elaborating the links between the sources identified and the reasons why they have been selected. Quotes are interesting, but they  appear listed in a mechanic way, without letting emerge a clear rationale for their choice and use.  

In a similar  vein the research questions/hypothesis as they are  presented now,  are repetitive and redundant: 14 sentences (7 + 7) starting in the same way...  and the sample is a mix of words and numbers complex to approach.

I think that the paper would benefit from a careful re-design with clear tables followed by friendly explanations. E.g. research hypothesis, sample, research instruments could be presented with a table then explained in words.

Some minor editing inaccuracies are present here and there (e.g. line 63 dot missing, 126 character etc.)

Author Response

Manuscript title:

A Study on Teachers’ Continuance Intention to Use Technology in English Instruction in Western China Junior Secondary Schools

Dear Reviewer,

Sub. Submission of Revised Paper " A Study on Teachers’ Continuance Intention to Use Technology in English Instruction in Western China Junior Secondary Schools"

   Article ID: sustainability-2192501

  Thank you for your email dated 30/01/2023 enclosing the reviewers' comments. We have carefully reviewed the comments and have revised the manuscript accordingly.  The language was edited by MDPI editing service. The responses are given in a point-by-point manner below.

S/N

Reviewer 3’s comments

Corrections by authors

1.

the paper offers the results of an interesting research tackling  important issues. However the way it is presented makes the reading difficult and should require some further efforts to better elaborate notions facts and results. In particular I think that the use of a large number of acronyms is confusing and results in a sort of treasure hunt of meanings. In some  cases words could be reported extensively rather than using acronyms and in any case a table of acronym disambiguation should be presented in order to allow the reader to have a quick reference when needed.

Abbreviations list was added and replacement of acronyms with their stands took part throughout the paper. However, we kept acronyms just in the figures:

 Abbreviations: EFL: English as a foreign language, HS: help seeking, IN: interest, ER: effort regulation, GM: growth mindset, FC: facilitating conditions, PU: perceived usefulness, PEU: perceives ease of use, SE: self-efficacy, CI: continuance intention

2.

the literature review can be improved, further elaborating the links between the sources identified and the reasons why they have been selected. Quotes are interesting, but they  appear listed in a mechanic way, without letting emerge a clear rationale for their choice and use. 

  The literature review was revised

3

In a similar  vein the research questions/hypothesis as they are  presented now,  are repetitive and redundant: 14 sentences (7 + 7) starting in the same way...  and the sample is a mix of words and numbers complex to approach.

The research hypotheses were rewritten as following to avoid repetition:

Ø  There is a direct effect of effort regulation (H1), Facilitating conditions (H2), interest (H3), growth mindset (H4), help seeking (H5), perceived ease of use (H6), perceived usefulness (H7) on teachers’ continuance intention to use technology in EFL instruction

Ø  There is an indirect effect of effort regulation (H8), Facilitating conditions (H9), interest (H10), growth mindset (H11), help seeking (H12), perceived ease of use (H13), perceived usefulness (H14) on teachers’ continuance intention to use technology in EFL instruction through self-efficacy

 The sample section was re-written and table of participants was added as following:

 The final sample of participants comprised 459 English language teachers from different regions in Western China, including Guangxi, Guizhou, Gansu, Qinghai, Xinjiang, Yunnan, Ningxia, Shaanxi, Enshi Autonomous Prefecture, and Hubei, during the academic year 2021-2022. There were sixteen junior secondary schools participated in this study. Most of these schools are based in rural and urban areas in Western China and they all use technology in instruction and learning. There were eight located in urban areas, whereas eight located in rural areas in Western China. As shown in table (1), of the 459 teachers, 216 were male and 243 were female; 206 teachers have Bachelor degree, 163 teachers have Master degree, whereas 90 teachers have other degrees. The age of the teachers categorized to five categories: 69 teachers are 30 years old and less, 133 teachers are between 31 and 35, 82 teachers are between 36 and 40 years old, 105 teachers are between 41 and 45 years old, and 70 teachers are between 46 and above.

Table 1. Participants’ Characteristics

Demographic variables

Frequency

Percentage

M

SD

Gender

459

100

1.529

0.499

Male

216

47.1

Female

243

52.9

Education Level

459

100

1.747

0.763

Bachelor

206

44.9

Master

163

35.5

Others

90

19.6

Age

459

100

2.943

1.315

30 years and less

69

15

31-35

133

29

36-40

82

17.9

41-45

105

22.9

46 years and above

70

15.3

I think that the paper would benefit from a careful re-design with clear tables followed by friendly explanations. E.g. research hypothesis, sample, research instruments could be presented with a table then explained in words.

Tables of participants and questionnaire factors and items were added

Some minor editing inaccuracies are present here and there (e.g. line 63 dot missing, 126 character etc.)

Editing was performed with one of MDPI editing agencies and certificate was provided

Round 2

Reviewer 2 Report

Thank you for the revisions!

Author Response

You're welcome, We really appreciate your comments

Reviewer 3 Report

OK

minor language issues still  to amend

Author Response

Manuscript title:

A Study on Teachers’ Continuance Intention to Use Technology in English Instruction in Western China Junior Secondary Schools

Dear  Reviewer,

Sub. Submission of Revised Paper " A Study on Teachers’ Continuance Intention to Use Technology in English Instruction in Western China Junior Secondary Schools"

   Article ID: sustainability-2192501

  Thank you for your email dated 15/02/2023 enclosing the reviewers' comments. We have carefully reviewed the comments and have revised the manuscript accordingly. the revisions were marked by tracked changes. The responses are given in a point-by-point manner below.

S/N

Reviewer 3’s comments

Corrections by authors

1.        

minor language issues still  to amend

The language has been revised again in the main manuscript